# Reducing charge noise in quantum dots by using thin silicon quantum wells

Brian Paquelet Wuetz[1], Davide Degli Esposti[1], Anne-Marije J. Zwerver[1], Sergey V. Amitonov[1,2], Marc Botifoll [3], Jordi Arbiol [3,4], Amir Sammak[2], Lieven M. K. Vandersypen [1], Maximilian Russ [1] & Giordano Scappucci [1]✉

Charge noise in the host semiconductor degrades the performance of spin-qubits and poses an obstacle to control large quantum processors. However, it is challenging to engineer the heterogeneous material stack of gate-defined quantum dots to improve charge noise systematically. Here, we address the semiconductor-dielectric interface and the buried quantum well of a $^{28}$Si/SiGe heterostructure and show the connection between charge noise, measured locally in quantum dots, and global disorder in the host semiconductor, measured with macroscopic Hall bars. In 5 nm thick $^{28}$Si quantum wells, we find that improvements in the scattering properties and uniformity of the two-dimensional electron gas over a 100 mm wafer correspond to a significant reduction in charge noise, with a minimum value of $0.29 \pm 0.02$ μeV/Hz$^{1/2}$ at 1 Hz averaged over several quantum dots. We extrapolate the measured charge noise to simulated dephasing times to cz-gate fidelities that improve nearly one order of magnitude. These results point to a clean and quiet crystalline environment for integrating long-lived and high-fidelity spin qubits into a larger system.

Spin-qubits in silicon quantum dots are a promising platform for building a scalable quantum processor because they have a small footprint[1], long coherence times[2,3], and are compatible with advanced semiconductor manufacturing[4]. Furthermore, rudimentary quantum algorithms have been executed[5] and quantum logic at high-fidelity performed[6–9]. As the qubit count is increasing, with a six-qubit processor demonstrated[10], significant steps have been taken to couple silicon spin qubits at a distance, via microwave photons or spin shuttling[11–16], towards networked spin-qubit tiles[17]. However, electrical fluctuations associated with charge noise in the host semiconductor can decrease qubit readout and control fidelity[18]. Reducing charge noise independently of the device location on a wafer is pivotal to achieving the ubiquitous high-fidelity of quantum operations, within and across qubit tiles, necessary to execute more complex quantum algorithms.

Charge noise is commonly associated with two-level fluctuators (TLF)[19] in the semiconductor host. In gated heterostructures with buried quantum wells, TLF may arise from impurities in several locations: within the quantum well, the semiconductor barrier, the semiconductor/dielectric interface, and the dielectrics layers above[20–26]. Furthermore, previous work on strained-Si MOSFETs[27–29], with strained-Si channels deposited on SiGe strain relaxed buffers, has associated charge noise with dislocations arising from strain relaxation, either deep in the SiGe buffer or at the quantum well/buffer interface. Since these impurities and dislocations are randomly distributed over the wafer and are also a main scattering source for electron transport in buried quantum wells[30], a holistic approach to materials engineering should be taken to address disorder in two-dimensional electron gases and charge noise in quantum dots.

[1]QuTech and Kavli Institute of Nanoscience, Delft University of Technology, PO Box 5046, 2600 GA Delft, The Netherlands. [2]QuTech and Netherlands Organisation for Applied Scientific Research (TNO), Delft, The Netherlands. [3]Catalan Institute of Nanoscience and Nanotechnology (ICN2), CSIC and BIST, Campus UAB, Bellaterra, 08193 Barcelona, Catalonia, Spain. [4]ICREA, Pg. Lluís Companys 23, 08020 Barcelona, Catalonia, Spain. ✉e-mail: g.scappucci@tudelft.nl

In this work, we demonstrate thin quantum wells in $^{28}$Si/SiGe heterostructures with low and uniform charge noise, measured over several gate-defined quantum dot devices. By linking charge noise measurements to the scattering properties of the two-dimensional electron gas, we show that a quiet environment for quantum dots is obtained by improving the semiconductor/dielectric interface and the crystalline quality of the quantum well. We feed the measured charge noise into a theoretical model, benchmark the model against recent experimental results[6,10], and predict that these optimized heterostructures may support long-lived and high-fidelity spin qubits.

## Results

### Description of $^{28}$Si/SiGe heterostructures

Figure 1a illustrates the undoped $^{28}$Si/SiGe heterostructures, grown by reduced-pressure chemical vapor deposition, and the gate-stack above. From bottom to top, the material stack comprises a 100 mm Si substrate, a strain-relaxed SiGe buffer layer, a strained $^{28}$Si quantum well, a 30 nm thick SiGe barrier, a Si cap oxidized in air to form a $SiO_x$ layer, an $AlO_x$ layer formed by atomic layer deposition, and metallic gates. The SiGe layers above and below the quantum well have a Ge concentration of $\simeq 0.3$ (Methods).

We consider three $^{28}$Si/SiGe heterostructures (A, B, C) to improve, in sequence, the semiconductor/dielectric interface (from A to B) and the crystalline quality of the quantum well (from B to C).

Heterostructure A has an $\simeq 9$ nm thick quantum well and is terminated with an epitaxial Si cap grown by dichlorosilane at 675 °C. This kind of heterostructure has already produced high performance spin-qubits[6,10,31]. Heterostructure B misses a final epitaxial Si cap but features an amorphous Si-rich layer obtained by exposing the SiGe barrier to dichlorosilane at 500 °C. Compared to A, heterostructure B supports a two-dimensional electron gas with enhanced and more uniform transport properties across a 100 mm wafer, owing to a more uniform $SiO_x$ layer with less scattering centers[32]. Finally, we introduce here heterostructure C, having the same amorphous Si-rich termination as in heterostructure B, but a thinner quantum well of $\simeq 5$ nm (Supplementary Fig. 1). This is much thinner than the Matthews-Blakeslee critical thickness[33,34], which is $\simeq 10$ nm[35] for the relaxation of tensile Si on $Si_{0.7}Ge_{0.3}$ via the formation of misfit dislocation at the bottom interface of the quantum well. In light of recent morphological characterization by electron channeling contrast imaging of Si/SiGe heterostructures with similar quantum well thickness and SiGe chemical composition[36], we expect misfit dislocation segments in heterostructure B because the quantum well approaches the Matthews-Blakeslee critical thickness. Due to the much thinner quantum well, instead, the epitaxial planes may adapt to the SiGe buffer much better in heterostructure C than in heterostructure B, meaning that misfit dislocations are, in principle, suppressed.

Figure 1b, c shows bright-field scanning transmission electron microscopy (BF-STEM) images from heterostructure C after

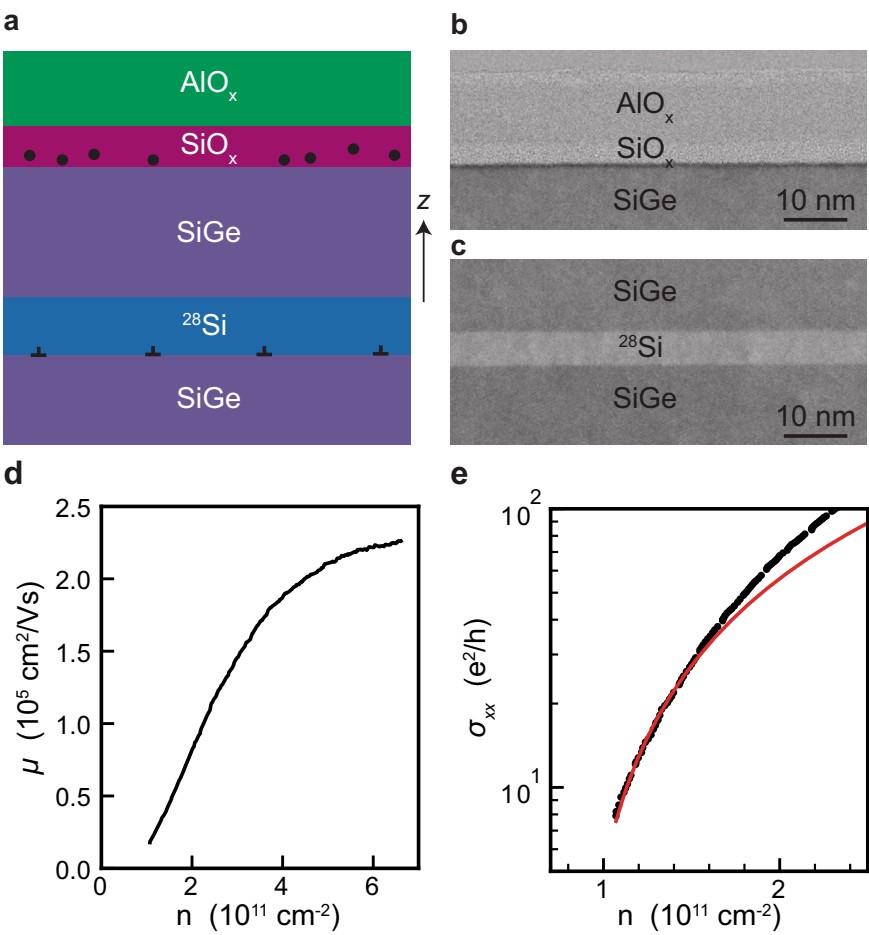

**Fig. 1 | Material stack and heterostructure field effect transistor characterization. a** Schematics of the $^{28}$Si/SiGe heterostructure and dielectric stack above. $z$ indicates the heterostructure growth direction. Circles represent remote impurities at the semiconductor/dielectric interface and perpendicular symbols represent misfit dislocations that might arise at the quantum well/buffer interface due to

strain relaxation. **b, c** BF-STEM images from heterostructure C highlighting the semiconductor/dielectric interface and the 5 nm thick $^{28}$Si quantum well, respectively. **d** Mobility $\mu$ and **e** conductivity $\sigma_{xx}$ measured as a function of density $n$ at a temperature of 1.6 K in a Hall bar H-FET from heterostructure C. The red curve in **e** is a fit to percolation theory.

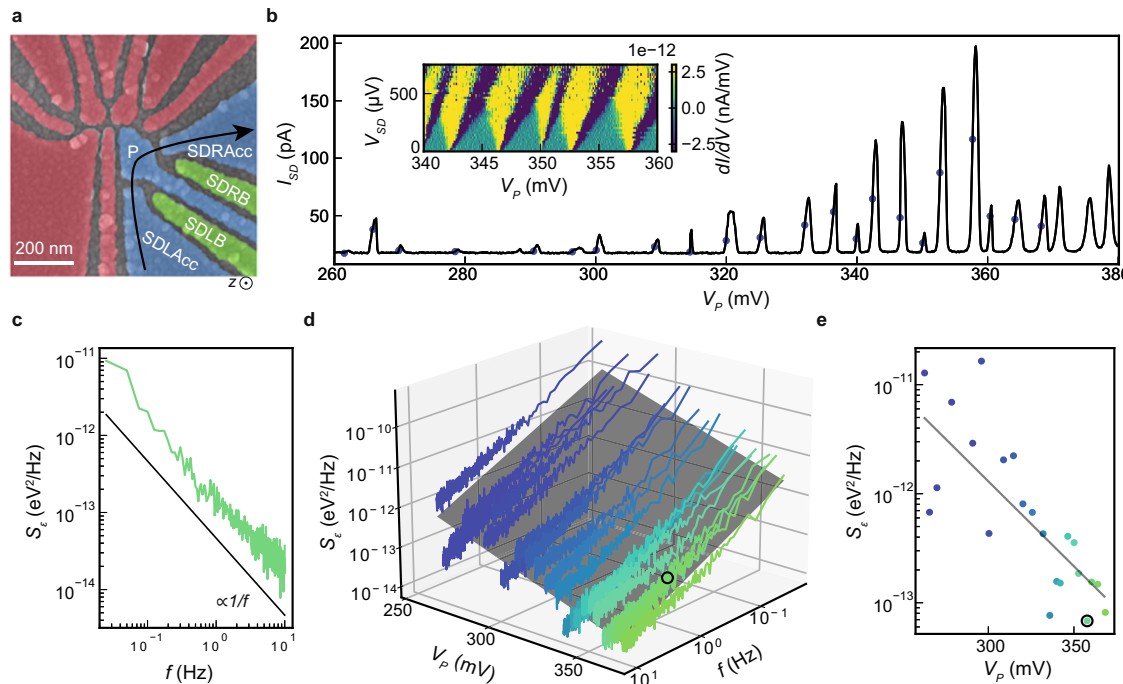

**Fig. 2 | Quantum dots and charge noise measurements. a** False colored SEM-image of a double quantum dot system with a nearby charge sensor. Charge noise is measured in the multi-electron quantum dot defined by accumulation gates SDLAcc and SDRAcc (blue), plunger P (blue), with the current going along the black arrow. In these experiments, the gates defining the double quantum dot (red) are used as screening gates. There is an additional global top gate (not shown) to facilitate charge accumulation when needed. **b** Source-drain current $I_{SD}$ through a charge sensor device fabricated on heterostructure C against the plunger gate voltage $V_P$. Colored dots mark the position of the flank of the Coulomb peak where charge noise measurements are performed. The inset shows Coulomb diamonds from the same device, plotted as the differential of the current $dI/dV$ as a function of $V_P$ and the source drain bias $V_{SD}$. **c** Charge noise spectrum $S_\epsilon$ measured at the Coulomb peak at $V_P \simeq 360.3$ mV in **b** and extracted using the lever arm from the corresponding Coulomb diamond. The black trendline is proportional to $1/f$. **d** $S_\epsilon$ for the same device in **b**, plotted in 3D as a function of $f$ and $V_P$. The dark gray plane is a fit through the datasets, i.e. the collection of noise spectra as in **c** measured at different $V_P$ and each obtained using a unique lever arm from the corresponding Coulomb diamond. **e** Line cut through the data in **d** at $f = 1$ Hz, showing the experimental noise $S_\epsilon$ (colored dots) and fit (dark gray line). The black circled data point (also in **d**) marks the minimum charge noise measured for this specific device ($S_{\epsilon,min}$) at $f = 1$ Hz.

fabrication of a Hall bar shaped heterostructure field effect transistors (H-FET). We observe a sharp SiGe/SiO$_x$ semiconductor/dielectric interface (Fig. 1b), characterized by a minor Ge pile up (dark line) in line with ref. [32]. The $\simeq 5$ nm thick quantum well (Fig. 1c, Supplementary Fig. 1) is uniform, has sharp interfaces to the nearby SiGe, and appears of high crystalline quality.

## Electrical characterization of heterostructure field effect transistors

We evaluate the scattering properties of the two-dimensional electron gases by wafer-scale electrical transport measured on Hall-bar shaped H-FETs operated in accumulation mode (Methods). For each heterostructure, multiple H-FETs over a wafer are measured in the same cooldown at a temperature of 1.7 K in refrigerators equipped with cryo-multiplexers[37]. Figure 1d, e shows typical mobility-density and conductivity-density curves for heterostructure C, from which we extract the mobility measured at high density ($n = 6 \times 10^{11}$ cm$^{-2}$) and the percolation density ($n_p$)[38]. The mobility rises steeply at low density due to progressive screening of scattering from remote impurities and flattens at higher density ($n > 5 \times 10^{11}$ cm$^{-2}$), limited by scattering from impurities within or nearby the quantum well, for example uniform background charges, surface roughness, or crystalline defects such as threading or misfit dislocations[30,39].

## Charge noise measurements in quantum dots

For charge noise measurements, we use devices comprising a double quantum dot and a charge sensor quantum dot nearby, illustrated in Fig. 2a. Using the same device design, two-qubit gates with fidelity

above 99% were demonstrated[6], silicon quantum circuits were controlled by CMOS-based cryogenic electronics[31], and energy splittings in $^{28}$Si/SiGe heterostructures were studied with statistical significance[40].

Here, we electrostatically define a multi-electron quantum dot in the charge sensor by applying gate voltages to the accumulation gates SDRAcc and SDLAcc, the barriers SDLB and SDRB, and the plunger gate P. All other gates (red in Fig. 2a) are set to 0 V for measurements of heterostructure B and C, whereas they are positively biased in heterostructure A to facilitate charge accumulation in the sensor (Methods). Figure 2b shows typical Coulomb blockade oscillations of the source-drain current $I_{SD}$ for a charge sensor from heterostructure C measured at a dilution refrigerator base temperature of 50 mK. We follow the same tune-up procedure (Methods) consistently for all devices and we measure charge noise at the flank of each Coulomb peak within the $V_P$ range defined by the first peak observable in transport and the last one before onset of a background channel (Supplementary Figs. 2–4). For example, in Fig. 2b we consider Coulomb peaks within the $V_P$ range from 260 mV to 370 mV. The data collected in this systematic way is taken as a basis for comparison between the three different heterostructures in this study.

For each charge noise measurement at a given $V_P$ we acquire 60 s (heterostructure A) or 600 s (heterostructures B, C) long traces of $I_{SD}$ and split them into 10 (heterostructure A) or 15 windows (heterostructures B, C). We obtain the current noise spectrum $S_I$ by averaging over the 10 (15) windows the discrete Fourier transform of the segments (Methods). We convert $S_I$ to a charge noise spectrum $S_\epsilon$ using, for each measurement at a given $V_P$, the unique lever arm

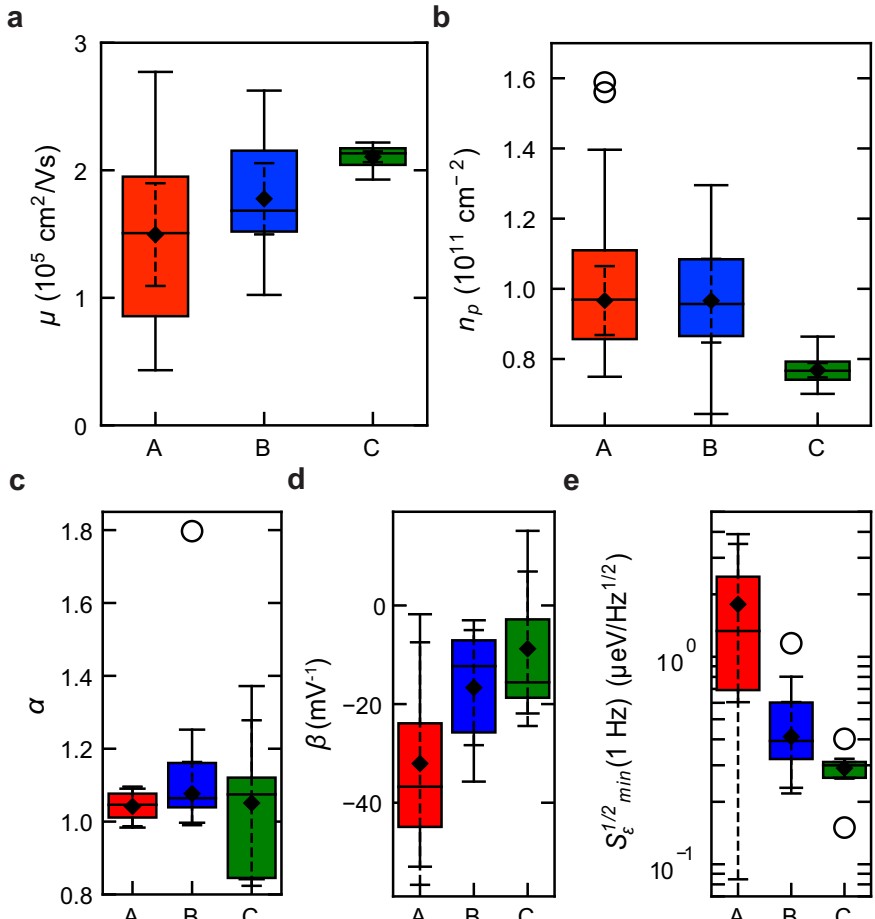

**Fig. 3 | Distribution of transport properties and charge noise. a, b** Distributions of mobility $\mu$ measured at $n = 6 \times 10^{11}$ cm$^{-2}$ and percolation density $n_p$ for heterostructure A (red, 20 H-FETs measured, of which 16 reported in ref. [32]), B (blue, 16 H-FETs measured of which 14 reported in ref. [32]), and C (green, 22 H-FETs measured). **c–e** Distributions of noise spectrum power law exponent $\alpha$, coefficient $\beta$ indicating the change in noise spectrum with increasing $V_P$, and minimum charge noise $S_{\epsilon,min}^{1/2}$ within the range of $V_P$ investigated for heterostructure A (red, 4 devices measured), B (blue, 7 devices measured), and C (green, 5 devices measured). Quartile box plots, mode (horizontal line), means (diamonds), 99% confidence intervals of the mean (dashed whiskers), and outliers (circles) are shown.

from the corresponding Coulomb diamonds and slope of the Coulomb peak to take into account a possible deformation of the charge sensor with the increasing electron number (inset Fig. 2b, Methods, and Supplementary Fig. 5). A representative charge noise spectrum $S_\epsilon$ measured at $V_P = 360.3$ mV is shown in Fig. 2c. We observe an approximate $1/f$ trend at low frequency, pointing towards an ensemble of TLF with a broad range of activation energies affecting charge noise around the charge sensor[41,42]. Figure 2e shows the charge noise $S_\epsilon$ at 1 Hz as a function of $V_P$. The charge noise decreases, with a linear trend, with increasing $V_P$, suggesting that, similar to scattering in 2D, screening by an increased electron density shields the electronically active region from noise arising from the heterostructure and the gate stack[43]. From this measurement we extract, for a given device, the minimum measured charge noise at 1 Hz ($S_{\epsilon,min}$ circled data point in Fig. 2e) upon variation of $V_P$ in our experimental range. We use $S_{\epsilon,min}$ as an informative metric to compare charge noise levels from device to device in a given heterostructure. For a given device, all charge noise spectra $S_\epsilon$ are plotted in 3D as a function of $f$ and $V_P$ (Fig. 2d). To quantify our observations, we fit the data to the plane $\log S_\epsilon = -\alpha \log f + \beta V_P + \gamma$ (Supplementary Note 4). Coefficient $\alpha = 0.84 \pm 0.01$ indicates the spectrum power law exponent and coefficient $\beta = -15.6 \pm 0.1$ mV$^{-1}$ quantifies the change in noise spectrum with increasing plunger gate and, consequently, the susceptibility of charge noise to the increasing electron number in the sensor.

## Distribution of transport properties and charge noise

We have introduced key metrics for 2D electrical transport ($\mu$, $n_p$) and charge noise ($\alpha$, $\beta$ and $S_{\epsilon,min}$) from Hall bar and quantum dot measurements, respectively. In Fig. 3a–e we compare the distributions of all these metrics for the three heterostructures A, B, C. Each box-plot is obtained from the analysis of measurements in Figs. 1d, e, and 2d repeated on multiple H-FETs or quantum dots, on dies randomly selected from different locations across the 100 mm wafers (Methods). To facilitate a comparison with previous studies, the minimum charge noise at 1 Hz is plotted in Fig. 3e as $S_{\epsilon,min}^{1/2}$ and therefore in units of μeV/Hz$^{1/2}$.

As reported earlier in ref. [32], the improvement in both mean values and spread for $\mu$ and $n_p$ was associated with a reduction of remote impurities when replacing the epitaxial Si cap in heterostructure A with a Si-rich passivation layer in heterostructure B. Moving to heterostructure C, we measure a high mean mobility of $(2.10 \pm 0.08) \times 10^5$ cm²/Vs and a low mean percolation density of $(7.68 \pm 0.37) \times 10^{10}$ cm$^{-2}$, representing an improvement by a factor $\simeq$1.4 and $\simeq$1.3, respectively (compared to heterostructure A). Most strikingly, the 99% confidence intervals of the mean for $\mu$ and $n_p$ are drastically reduced by a factor $\simeq$9.8 and $\simeq$4.8, respectively. We speculate that these improvements in heterostructure C are associated with the suppression of misfit dislocations at the quantum well/buffer interface, thereby reducing short range scattering and increasing uniformity on a wafer-scale. This interpretation is supported by previous

studies of mobility limiting mechanisms as a function of the quantum well thickness in strained Si/SiGe heterostructures[39]. We speculate that further reducing the quantum well thickness could increase surface roughness scattering from the bottom interface, and therefore disorder. Instead, fine-tuning the quantum well thickness between 5 nm and 9 nm might minimize surface roughness scattering whilst still avoiding the formation of misfit dislocations.

We now shift our attention to the results of charge noise measurements. First, the power law exponent $\alpha$ (Fig. 3c) shows a mean value $\simeq 1$, however the 99% confidence interval and interquartile range increase when moving from heterostructure A to B and C. Next, we observe a decreasing trend for the absolute mean value of coefficient $\beta$ (Fig. 3d), meaning that the noise spectrum is less susceptible to changes in $V_P$. Finally, Fig. 3e shows the distributions for $S_{\epsilon,min}^{1/2}$, the minimum charge noise at 1 Hz upon varying $V_P$. We find in heterostructure C an almost order of magnitude reduction in mean $S_{\epsilon,min}^{1/2}$ to $0.29 \pm 0.02$ µeV/Hz$^{1/2}$. This trend is confirmed by plotting the distributions of maximum charge noise at 1 Hz upon varying $V_P$ (Supplementary Fig. 4). Furthermore, within the distribution of $S_{\epsilon,min}^{1/2}$ for heterostructure C, the minimum value of the measured charge noise as a function of $V_P$ and across quantum dots is 0.15 µeV/Hz$^{1/2}$. These charge noise values are on par or compare favorably to the best values reported previously at 1 Hz in gate defined quantum dots. In multi-electron quantum dots, charge noise of 0.47 µeV/Hz$^{1/2}$ was reported for Si/SiGe[44], 0.6 µeV/Hz$^{1/2}$ (average value, with a minimum of $\leq$0.2 µeV/Hz$^{1/2}$) for Ge/SiGe[45], $0.49 \pm 0.1$ µeV/Hz$^{1/2}$ for Si/SiO$_2$[46], and 1 µeV/Hz$^{1/2}$ for InSb[47]. In single-electron quantum dots, charge noise of 0.33 µeV/Hz$^{1/2}$ was reported for Si/SiGe[48] and 7.5 µeV/Hz$^{1/2}$ for GaAs[49].

We understand the charge noise trends in Fig. 3c–e by relating them to the evolution of the disorder landscape moving from heterostructures A to B and C, as inferred by the electrical transport measurements in Fig. 3a, b. The narrow distribution of $\alpha$ in heterostructure A points to charge noise being dominated from many TLFs possibly located at the low quality semiconductor/dielectric interface and above, albeit other sources of charge noise in the surrounding environment of the quantum dot may be present, such as highly localized misfit dislocations arising from partial strain relaxation in the quantum well or other nearby fluctuators. With a better semiconductor/dielectric interface, the effect of these other nearby fluctuators emerges in heterostructure B and C as a larger spread of the frequency exponent $\alpha$, indicating a nonuniform distribution of activation energies according to the Dutta-Horn model[50]. Yet, the noise spectra still follow a 1/$f$-like behavior (Supplementary Fig. 3), suggesting that TLFs also experience slow temperature fluctuations[42]. The electrical transport measurements support this interpretation: scattering from many remote impurities is dominant in heterostructure A, whereas with a better semiconductor/dielectric interface remote scattering has less impact in the transport metrics of heterostructures B and C.

The decreasing trend in $|\beta|$ is in line with the observation from electrical transport. As the impurity density decreases from heterostructure A to B and C, charge noise is less affected by an increasing $V_P$, since screening of electrical noise through adding electrons to the charge sensor becomes less effective. While we are not able to measure directly the electron number in the charge sensor, we deem unlikely the hypothesis that charge sensors in heterostructure A are operated with considerably fewer electrons than in heterostructure C. This is because all operation gate voltages in heterostructure A are consistently larger than in heterostructure C (Supplementary Fig. 4), due to the higher disorder.

Finally, the drastic reduction in mean value and spread of $S_{\epsilon,min}^{1/2}$ mirrors the evolution of mean value and spread of $n_p$ and $\mu$. From heterostructure A to B, a reduction in scattering from remote impurities is likely to result in less charge noise from long-range TLFs. From heterostructure B to C, the reduction in the possible number of

dislocations at the quantum well/buffer interface, further reduces the charge noise picked up by quantum dots. This explanation is based on earlier studies of charge noise in strained Si-MOSFETs[27–29], which showed a correlation between low-frequency noise spectral density and static device parameters. Dislocations at the bottom of the strained channel may act as scattering centers that degrade mobility and as traps for the capture and release of carriers, which causes noise similarly to traps at the dielectric interface.

## Calculated dephasing time and infidelity

To emphasize the improvement of the electrical environment in the semiconductor host, we calculate the dephasing time $T_2^\star$ of charge and spin qubits assuming these qubits experience the same fluctuations as our $^{28}$Si/SiGe quantum dots. The dephasing time of a qubit (in the quasistatic limit and far-off from a sweet spot) is given by[51,52]

$$T_2^\star = \frac{h}{\sqrt{2\pi}\sigma} \qquad (1)$$

with the Planck constant $h$ and the standard deviation

$$\sigma^2 = \left|\frac{\partial \mathcal{E}}{\partial \mu}\right|^2 \times 2 \int_{f_{low}}^{f_{high}} \frac{S_\epsilon^2}{f^\alpha} df. \qquad (2)$$

Importantly, both the charge noise amplitude $S_\epsilon^2(f)$ and the noise exponent $\alpha$ have a strong impact on the dephasing time while the low and high frequency cut-off, $f_{low}$ and $f_{high}$, given by the duration of the experiment have a weaker impact. The prefactor $|\frac{\partial \mathcal{E}}{\partial \mu}|$ translates shifts in chemical potential of the charge sensor into energy shifts of the qubit and depends on many parameters such as the type of qubit and the device itself. We find $|\frac{\partial \mathcal{E}}{\partial \mu}| = 1$ for a charge qubit[53] and $|\frac{\partial \mathcal{E}}{\partial \mu}| \approx 10^{-5}$ for an uncoupled spin- qubit[44] (see Supplementary Note 7 for a derivation of these numbers and the used frequency bandwidths).

Figure 4a shows the computed dephasing times of charge qubits (circle) and spin qubits (star) for all three heterostructures. These calculations represent a best case scenario, since we use the distribution of measured $S_{\epsilon,min}$ from Fig. 3 as input parameter for each heterostructure. The improvements in our material can be best seen by investigating $T_2^\star$ of the charge qubit since it is directly affected by charge noise. Our theoretical extrapolation shows two orders of magnitude improvement in $T_2^\star$ by switching from heterostructures A to heterostructures B and C. One order is gained from the reduced charge noise amplitude and another order is gained through a more beneficial noise exponent $\alpha > 1$. Note, that the integration regimes differ for spin and charge qubits due to the different experimental setups and operation speeds[44,53]. For potential spin qubits in heterostructure A the calculated $T_2^\star$ shows an average $\overline{T}_2^\star = 8.4 \pm 5.6$ µs. This distribution compares well with the distribution $\overline{T}_2^\star = 6.7 \pm 5.6$ µs of experimental $T_2^\star$ data from state-of-the-art semiconductor spin qubits in materials with similar stacks as in heterostructure A[6,10]. Note that while such comparisons oversimplify actual semiconductor spin-qubit devices by reducing them to a single number, they fulfill two aims. They allow us to benchmark the computed performance of heterostructure A to past experiments and provide a prognosis on the qubit quality in novel material stacks. Heterostructures B and C, in this case, may support average dephasing times of $\overline{T}_2^\star = 24.3 \pm 12.5$ µs and $\overline{T}_2^\star = 36.7 \pm 18$ µs, respectively. The highest values $T_2^\star = 70.1$ µs hints towards a possible long dephasing time for spin qubits, previously only reported in ref. [2].

Figure 4b shows the simulated infidelity, a metric to measure the closeness to the ideal operation, of a universal cz-gate between two spin qubits following ref. [6] and Supplementary Note 7. Note that the device used in ref. [6] has the same architecture as our test devices. In the cz-gate simulation, noise couples in dominantly via barrier

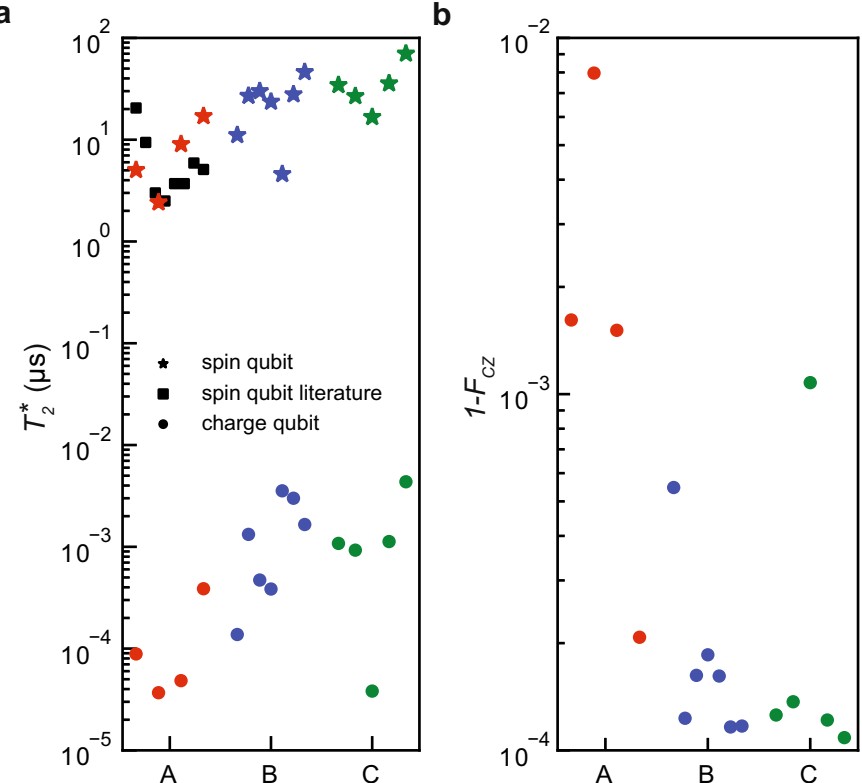

**Fig. 4 | Calculated dephasing times and infidelity. a** Computed dephasing times $T_2^*$ of a charge qubit (circle) and of a spin-qubit (star) using $S_{\epsilon,min}$ from heterostructure A (red), B (blue), C (green). Eq. (1) was used to compute $T_2^*$ as a function of $S_\epsilon$ and $\alpha$ from Fig. 3 with frequency cutoffs $(f_{min}, f_{max}) = (1.6\text{ mHz}, 33\text{ GHz})$ and $(f_{min}, f_{max}) = (1.6\text{ mHz}, 10\text{ kHz})$. Literature values (squares) are taken from refs. [6,10]. **b** Simulated infidelity of a cz-gate between two spin qubits following the ref. [6] using $S_\epsilon$ and $\alpha$ from heterostructure A (red), B (blue), C (green) in Fig. 3 as input for barrier fluctuations.

voltage fluctuations which affects the interaction between the electron spins. Again, we use the charge noise amplitude $S_{\epsilon,min}$ and exponent $\alpha$ from the quantum dot experiments in Fig. 3 as input for the simulations. The simulations show an averaged average gate infidelity $1 - \bar{F}_{\text{CZ}} = 0.02 \pm 0.01\%$ which means on average a single error every 5000 runs. We also observe a saturation value close to $1 - F = 10^{-4}$ which arises from single-qubit dephasing $T_2^* = 20\ \mu s$ used in the simulations estimated from nuclear spin noise due to a 800 ppm concentration of the $^{29}$Si silicon isotope which has a non-zero nuclear spin[44].

## Discussion

In summary, we have measured electron transport and charge noise in $^{28}$Si/SiGe heterostructures where we improve the semiconductor/dielectric interface, by adopting an amorphous Si-rich passivation, and the structural quality of the quantum well, by reducing the quantum well thickness significantly below the Matthew-Blakeslee critical thickness for strain relaxation. We relate disorder in 2D to charge noise in quantum dots by following a statistical approach to measurements. A reduction of remote impurities and dislocations nearby the quantum well is connected with the key improvements in the scattering properties of the 2D electron gas, such as mobility and percolation density, and their uniformity across a 100 mm wafer. The trend observed from electron transport in 2D is compatible with the observations from measurements of charge noise in quantum dots. As remote impurities are reduced, charge noise becomes more sensitive to local fluctuators nearby the quantum well and less subject to screening by an increased number of electrons in the dot. Furthermore, with this materials optimization, we achieve a statistical improvement of nearly one order of magnitude in the charge noise supported by quantum dots. Using the charge noise distribution as input parameter and benchmarking

against published spin-qubit data, we predict that our optimized semiconductor host could support long-lived and high-fidelity spin qubits. We envisage that further materials improvements in the structural quality of the quantum well, in addition to the commonly considered semiconductor/dielectric interface, may lead systematically to quantum dots with less noise and to better qubit performance.

## Methods

### Si/SiGe heterostructure growth

The $^{28}$Si/SiGe heterostructures are grown on a 100-mm n-type Si(001) substrate using an Epsilon 2000 (ASMI) reduced pressure chemical vapor deposition reactor. The reactor is equipped with a $^{28}$SiH$_4$ gas cylinder (1% dilution in H$_2$) for the growth of isotopically enriched $^{28}$Si. The $^{28}$SiH$_4$ gas was obtained by reducing $^{28}$SiF$_4$ with a residual $^{29}$Si concentration of 0.08%[54]. Starting from the Si substrate, the layer sequence of all heterostructures comprises a 3 μm step-graded Si$_{(1-x)}$Ge$_x$ layer with a final Ge concentration of $x = 0.3$ achieved in four grading steps ($x = 0.07$, 0.14, 0.21, and 0.3), followed by a 2.4 μm Si$_{0.7}$Ge$_{0.3}$ strain-relaxed buffer. The heterostructures differ for the active layers on top of the strain-relaxed buffer. Heterostructure A has a 9 nm tensile strained $^{28}$Si quantum well, a 30 nm Si$_{0.7}$Ge$_{0.3}$ barrier, and a sacrificial 1 nm epitaxial Si cap. Heterostructure B has an 9 nm tensile strained $^{28}$Si quantum well, a 30 nm Si$_{0.7}$Ge$_{0.3}$ barrier, and a sacrificial passivated Si cap grown at 500 °C. Heterostructure C has a 5 nm tensile strained $^{28}$Si quantum well, a 30 nm Si$_{0.7}$Ge$_{0.3}$ barrier, and a sacrificial passivated Si cap grown at 500 °C. A typical secondary ions mass spectrometry of our heterostructures is reported in Supplementary Fig. S13 of ref. [40] and the Ge concentration in the SiGe layers is confirmed by quantitative electron energy loss spectroscopy (EELS).

## Device fabrication

The fabrication process for Hall-bar shaped heterostructure field effect transistors (H-FETs) involves: reactive ion etching of mesa-trench to isolate the two-dimensional electron gas; P-ion implantation and activation by rapid thermal annealing at 700 °C; atomic layer deposition of a 10-nm-thick $Al_2O_3$ gate oxide; deposition of thick dielectric pads to protect gate oxide during subsequent wire bonding step; sputtering of Al gate; electron beam evaporation of Ti:Pt to create ohmic contacts to the two-dimensional electron gas via doped areas. All patterning is done by optical lithography. Double quantum dot devices are fabricated on wafer coupons from the same H-FET fabrication run and share the process steps listed above. Double-quantum dot devices feature a single layer gate metallization and further require electron beam lithography, evaporation of Al (27 nm) or Ti:Pd (3:17 nm) thin film metal gate, lift-off, ALD of a 5 nm thick $Al_2O_3$ insulating layer, and a global top-gate.

## Electrical characterization of H-FETs

Hall-bar H-FETs measurements are performed in an attoDRY2100 variable temperature insert refrigerator at a base temperature of 1.7 K[32]. We apply a source-drain bias of 100 µV and measure the source-drain current $I_{SD}$, the longitudinal voltage $V_{xx}$, and the transverse Hall voltage $V_{xy}$ as function of the top gate voltage $V_g$ and the external perpendicular magnetic field $B$. From here we calculate the longitudinal resistivity $\rho_{xx}$ and transverse Hall resistivity $\rho_{xy}$. The Hall electron density $n$ is obtained from the linear relationship $\rho_{xy} = B/en$ at low magnetic fields. The carrier mobility $\mu$ is extracted from the relationship $\sigma_{xx} = ne\mu$, where $e$ is the electron charge. The percolation density $n_p$ is extracted by fitting the longitudinal conductivity $\sigma_{xx}$ to the relation $\sigma_{xx} \propto (n - n_p)^{1.31}$. Here $\sigma_{xx}$ is obtained via tensor inversion of $\rho_{xx}$ at $B = 0$. The box plots in Fig. 3a, b for heterostructure A (red) and B (blue) expand previously published data in Fig. 2f, e of ref. [32] by considering measurements of 4 additional H-FETs for heterostructure A (20 H-FETs in total) and of 2 additional H-FETs for heterostructure B (16 H-FETs in total).

## Electrical characterization of quantum dots

Measurements of the multi-electron quantum dots defined in the charge sensor are performed in a Leiden cryogenic dilution refrigerator with a mixing chamber base temperature $T_{MC} = 50$ mK[40]. The devices are tuned systematically with the following procedure. We sweep all gate voltages ($V_{SDRAcc}$, $V_{SDRB}$, $V_P$, $V_{SDLB}$, and $V_{SDLAcc}$) from 0 V towards more positive bias, until a source-drain current $I_{SD}$ of ≈1 nA is measured, indicating that a conductive channel has formed in the device. We then reduce the barrier voltages to find the pinch-off voltages for each barrier. Subsequently, we measure $I_{SD}$ as a function of $V_{SDLB}$ and $V_{SDRB}$ and from this 2D map we find a set of gate voltage parameters so that Coulomb blockade peaks are visible. We then fix the barrier voltages and sweep $V_P$ to count how many clearly defined Coulomb peaks are observed before onset of a background current. The quantum dot is tuned to show at least 9 Coulomb peaks, so that noise spectra may be fitted as in Fig. 2d with meaningful error bars. If we see less than 9 Coulomb peaks we readjust the accumulation gate voltages $V_{SDRAcc}$, and $V_{SDLAcc}$, and repeat the 2D scan of $V_{SDLB}$ against $V_{SDRB}$. In one case (device 2 of heterostructure A), we tuned device to show past 5 Coulomb peaks and still performed the fit of the charge noise spectra similar to the one shown in Fig. 2d. Further details on the extraction of the lever arms and operation gate voltages of the devices are provided in Supplementary Figs 4 and 5. We estimate an electron temperature of 190 mK by fitting Coulomb blockade peaks (see Supplementary Fig. 2 in ref. [32]) measured on quantum dot devices.

For heterostructure A we apply a source drain bias of 100 µV (1 device) or 150 µV (3 devices) across the quantum dot, finite gate voltages across the operation gates of the dot, and finite gate voltages across the screening gates. We measure the current $I_{SD}$ and the current noise spectrum $S_I$ on the left side of the Coulomb peak where $|dI/dV_P|$ is largest. We use a sampling rate of 1 kHz for 1 min using a Keithley DMM6500 multimeter. The spectra are then divided into 10 segments of equal length and we use a Fourier transform to convert from time-domain to frequency-domain for a frequency range of 167 mHz–500 Hz. We set the upper limit of the frequency spectra at 10 Hz, to avoid influences from a broad peak at around 150 Hz coming from the setup (Supplementary Fig. 3). A peak in the power spectral density at 9 Hz is removed from the analysis since it is an artifact of the pre-amplifier. To convert the current noise spectrum to a charge noise spectrum, we use the formula[20]

$$S_\epsilon = \frac{a^2 S_I}{|dI/dV_P|^2} \tag{3}$$

where $a$ is the lever arm and $|dI/dV_P|$ is the slope of the Coulomb peak at the plunger voltage used to acquire the time trace.

The charge noise measurements conditions have been slightly modified from sample A to sample B, C to extend the probed frequency range from 100 µHz to 10 µHz. For heterostructures B and C we apply a source drain bias of 150 µV across the quantum dot, finite gate voltages across the operation gates of the quantum dot, and we apply 0 V to all other gates. We measure the current $I_{SD}$ and the current noise spectrum $S_I$ on the left side of the Coulomb peak where $|dI/dV_P|$ is largest. We use a sampling rate of 1 kHz for 10 min using a Keithley DMM6500 multimeter. The spectra are then divided into 15 segments of equal length and we use a Fourier transform to convert from time-domain to frequency-domain for a frequency range of 25 mHz–500 Hz. We set the upper limit of the frequency spectra at 10 Hz, to avoid influences from a broad peak at around 150 Hz coming from the setup. We use Eq. (3) to convert the current noise spectrum to a charge noise spectrum.

## (Scanning) Transmission Electron Microscopy

For structural characterization with (S)TEM, we prepared cross-sections of the quantum well heterostructures by using a Focused Ion Beam (Helios 600 dual beam microscope). Atomically resolved HAADF STEM data was acquired in a probe corrected TITAN microscope operated at 300 kV. Quantitative EELS was carried out in a TECNAI F20 microscope operated at 200 kV with approximately 2 eV energy resolution and 1 eV energy dispersion. Principal Component Analysis (PCA) was applied to the spectrum images to enhance S/N ratio.

## Data availability

All data included in this work are available from the 4TU.ResearchData international data repository at https://doi.org/10.4121/20418579.

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

## Acknowledgements

We acknowledge helpful discussions with G. Isella, D. Paul, M. Mehmandoost, the Scappucci group and the Vandersypen group. This research was supported by the European Union's Horizon 2020 research and innovation programme under the Grant Agreement No. 951852 (QLSI project) and in part by the Army Research Office (Grant No. W911NF-17-1-0274). The views and conclusions contained in this

document are those of the authors and should not be interpreted as representing the official policies, either expressed or implied, of the Army Research Office (ARO), or the U.S. Government. The U.S. Government is authorized to reproduce and distribute reprints for Government purposes notwithstanding any copyright notation herein. M.R. acknowledges support from the Netherlands Organization of Scientific Research (NWO) under Veni grant VI.Veni.212.223. ICN2 acknowledges funding from Generalitat de Catalunya 2021SGR00457. ICN2 is supported by the Severo Ochoa program from Spanish MCIN / AEI (Grant No.: CEX2021-001214-S) and is funded by the CERCA Programme / Generalitat de Catalunya and ERDF funds from EU. Part of the present work has been performed in the framework of Universitat Autònoma de Barcelona Materials Science PhD program. Authors acknowledge the use of instrumentation as well as the technical advice provided by the National Facility ELECMI ICTS, node "Laboratorio de Microscopias Avanzadas" at University of Zaragoza. M.B. acknowledges support from SUR Generalitat de Catalunya and the EU Social Fund; project ref. 2020 FI 00103. We acknowledge support from CSIC Interdisciplinary Thematic Platform (PTI+) on Quantum Technologies (PTI-QTEP+).

## Author contributions

A.S. grew and designed the 28Si/SiGe heterostructures with B.P.W. and G.S.. M.R. developed the theory. A.S. and D.D.E. fabricated heterostructure field effect transistors measured by B.P.W. and D.D.E.. M.B and J.A. performed TEM characterization. S.A and D.D.E. fabricated quantum dot devices. B.P.W. and D.D.E. measured the quantum dot devices with contributions from A.M.J.Z.. G.S. conceived and supervised the project. B.P.W, D.D.E, M.R, and G.S. wrote the manuscript with input from all authors.

## Competing interests

The authors declare no competing interests.
