## [Peer Review File · Nature Communications]

Reducing charge noise in quantum dots by using thin silicon quantum wellsREVIEWER COMMENTS

Reviewer #1 (Remarks to the Author):

This paper compares the effect on charge noise from two essential material parameters of SiGe heterostructures for electron spin qubits – either Si epitaxial cap or amorphous cap; the Si quantum well thickness. Though the interpretation of material properties is based on the well-established dislocation theory instead of experimental results, this study compares the charge noise between the samples carefully by electrical measurements, in this case, Hall and charge noise. This study is highly relevant to improve the electron spin qubits by optimising the SiGe heterostructures.

Therefore, I recommend the paper for publication. There are only several minor points to discuss here:

- 1) Only one parameter is mentioned in the title, that is the quantum well thickness. I recommend adding the other parameter also into the title, such as “.. by using thin silicon quantum well and amorphous Si cap”. It's only a suggestion.
- 2) It mentions in the paper that by using the amorphous Si cap can reduce impurities. May I ask which impurities you mean and why using amorphous Si can reduce the impurities?
- 3) “No structural defects such as misfit dislocation are visible suggesting they are at most scarce” – It is anyway hard to observe misfit dislocation in cross-view TEM since the magnification is too high and the view scale is very limited. Figure 1b is probably around 100 nm x 100 nm.
- 4) This study shows that reducing the quantum well thickness can reduce the charge noise in the quantum dots. Till how thin can the quantum well thickness be reduced?
- 5) There's a study [Lodari et al, Mat. Quant. Tech. (2021)] from the same group on the Ge hole qubit - by increasing the SiGe top buffer layer thickness, the charge noise is reduced. Will it be also the same case by Si quantum well for electron qubit? Can authors give a comment here?
- 6) Since I am personally not familiar with charge noise characterization – the charge characterization part makes sense to me, however, other experts to review this part is necessary. Therefore, I actually have several questions regards to this part:
 - a. The measurement conditions are slightly different from sample A to sample B, C – why?
 - b. Maybe add some example of the ISD sampling in the supplementary data – Plot of time-resolved ISD and related f-dependent SI.
 - c. The author said, “For each charge noise measurement at a given VP we acquire 60 s (heterostructure A) or 600 s (heterostructures B, C) long traces of ISD and split them into 10 (heterostructure A) or 15 windows (heterostructures B, C). We obtain the current noise spectrum SI by averaging over the 10 (15) windows the discrete Fourier transform of the segments (Methods).”. What's the meaning of windows here?
 - d. The equation $\log S_e = -\alpha \log f + \beta V_p + \gamma$. May I ask the source? Maybe can also be added into supplementary materials
- 7) Citation [50] is about superconducting qubits, and I didn't find the equation listed in the citation. There are some similar equations in citation [50] but not the same. Instead, the equation was found in [Bermeister, APL (2013)]. Again, since I am not an expert in charge noise, maybe it's my mistake.

– I think the strain relaxation measured by Raman shift needs further discussion also. Raman shift can differ from the same sample dependent on the measurement position. Therefore, only one plot from one measurement point from each sample to say one sample is more relaxed than another one makes not much sense. Besides, I don't think even the Si layer is beyond critical thickness and there's partial relaxation, but the relaxation can be up

to 0.4%. According to the calculation from Matthews-Blakeslee paper, the relaxation of Si with Misfit dislocation spacing of around 1 micron is around 0.02%, that is way smaller than the value observed in this paper.

Reviewer #2 (Remarks to the Author):

This manuscript provides a welcome study of charge noise levels in Si/SiGe quantum dots for different device designs. Of great importance, the authors present quantitative findings of reduced noise strength when using amorphous Si-rich passivation to reduce interface defects and, strikingly, an order of magnitude reduction in variability when also using thin quantum wells to reduce dislocation defects. This is accompanied with insightful discussion of the implications for the type and location of remaining noise sources. I cannot emphasize strongly enough that this manuscript squarely addresses the most pressing issue for semiconductor-based qubits and sheds important new light on it. These data will be invaluable for the field moving forward. The manuscript is well-written and the methodology is sound. Everything is completely appropriate for the high standards of Nature Communications.

I have a few technical comments:

- 1) On pg 3, it would be helpful to state why the specific numbers given for the noise trace lengths and number of windows were chosen, and in particular why they are not chosen to be the same for all heterostructures.
- 2) On pg 4, the units of β do not make sense to me. It cannot have the same units as S/V_P .
- 3) Fig. 2d, from heterostructure C, is said to be fit by the plane with $\alpha = 0.84$, but in Fig. 3c this value seems to be at the extreme lower end of the distribution. Did the data for Fig. 2d come from an atypical H-FET?
- 4) On pg 5, the authors write that "the larger spread in α in heterostructure B and C implies that deviations from $1/f$ behaviour become more frequent, possibly originating from a non-uniform distribution of TLF or from one low frequency TLF in the surrounding environment..." While I agree that a nonuniform distribution of TLF activation energies would cause deviations from $1/f$ behavior, it is not clear to me why one should expect that the remaining dislocations in B would have a less uniform distribution of activation energies than the interfacial impurities. Do the authors have an idea why this might be? Furthermore, if one simply has a dominant low-frequency TLF nearby, the power spectrum would be Lorentzian and would certainly not look like Fig 2 over nearly three decades of frequency. That is, unless that TLF also experiences slow temperature fluctuations, in which case Ref. 42 has shown that a single TLF *does* produce $1/f$ noise (intriguingly, with α also around 0.8). The authors should clarify this.
- 5) On pg 5, I agree that it makes sense to interpret the decreasing trend in $|\beta|$ as a reduction in the effectiveness of screening. That is consistent with the charge noise being increasingly dominated by nearby TLFs rather than remote scatterers. However, I do not know if that is what the authors have in mind when they mention a smaller "TLF-per-volume" ratio, and this phrase should be clarified.

In conclusion, I strongly recommend publication of this manuscript in Nature Communications. It presents vital clues for the biggest mystery in semiconductor-based quantum computing.

Reviewer #3 (Remarks to the Author):

Referee Report - NCOMMS-22-39786-T

Title: Reducing charge noise in quantum dots by using thin silicon quantum wells

Authors: B. Paquelet Wuetz,¹ D. Degli Esposti,¹ A.M.J. Zwerver,¹ S.V. Amitonov,^{1,2} M. Botifoll,³ J. Arbiol,^{3,4} A. Sammak,² L.M.K. Vandersypen,¹ M. Russ,¹ and G. Scappucci^{1,*}

Report:

In this manuscript, the authors report an extension of their expansive body of work using Si quantum wells in SiGe heterostructures to form quantum dots, in this instance by reducing the quantum well thickness while using an alternative gate dielectric approach reported previously. Here, the authors compare this new heterostructure with two previously published heterostructures using transport in H-FET (Hall Field Effect Transistors) devices to assess many-body conduction properties and quantum dot gate structures to examine stability in the single electron regime. Notably, the authors present measurements of time dependent fluctuations of the current through quantum dot devices, which they analyze in the context of charge noise, and extrapolate these results using a theoretical construction to imagine and compare qubit performance between the three heterostructures.

Overall, I believe this paper is a strong candidate for publication, but opportunities for strengthening the clarity, discussion and impact are offered below, which I encourage the authors to consider. The new heterostructure reported in this paper is likely to be of great interest to the quantum information community and the material changes made are likely to be replicated by others. This material system is seen by many to be a viable choice for large scale semiconductor quantum information, and variations of this kind are important to explore and report, whether of great or marginal benefit.

In the example determination of S_{E_min} presented in Fig. 2 and S3, the conversion from S_I to S_E depends on the lever arm and the slope of the Coulomb blockade peak ($|dV/dp|$), as defined in Eq. 3. Do the authors use a single value of the lever arm/slope for each device, or is the value uniquely determined for each peak? If the former, then some justification and discussion of this choice is appropriate. If the latter, I would welcome an additional supplemental figure which demonstrated for the case presented in Fig 2 the values of lever arm and slope found – not just for assessing the conversion of data to charge noise, but also for consideration in subsequent discussion of screening, etc. I would imagine this as similar to Fig S4, which shows ranges of other operating parameters.

Continuing from Fig 2d, where the derived S_E data for various CB peaks are plotted along with the “plane” used to capture the character of the device, where is S_{E_min} in this plot (Fig 2d) as determined by the authors? Please add a special symbol to explicitly show the S_{E_min} location in this plot. Additionally, if the same units were used in Figs 2 c – e, this would be much more helpful to the reader. Similarly for showing S_{E_min} in Fig 2e and connecting to the data in Fig S4.

While I recognize the main experimental part leverages comparison based on parameters for

the fit function (what happens with γ ?), the theoretical extrapolation to T_2^* and F are determined entirely by S_{E_min} . As presented, the data, extrapolations, and eventual conclusions can only be qualitatively compared since it is not clear how S_{E_min} is determined by the authors, and why that determination is a robust means of making comparisons between the devices, which have some variation in tuning, etc. Particularly for the choices in the theoretical section, and the choices of things like the high and low frequency cutoffs (which don't overlap with the data at all in one case?) additional discussion about the robustness for intercomparison and whether the S_{E_min} is robust if the same device was tuned up multiple times would help establish more confidence in this approach. Why should the reader trust S_{E_min} ?

Finally, in the theoretical section and supplemental, additional attention to wordsmithing and imprecise language could help with clarity.

Delft, February 2, 2023

1. RESPONSE TO REVIEWER 1

We thank the Reviewer for acknowledging the relevance of our work and for recommend the manuscript for publication. We have taken into account all the minor (and constructive) points raised by the Reviewer that we address below point by point.

1.1 Only one parameter is mentioned in the title, that is, the quantum well thickness. I recommend adding the other parameter also into the title, such as “. by using thin silicon quantum well and amorphous Si cap”. It’s only a suggestion.

Our reply: We thank the Reviewer for this suggestion. While the critical role of the semiconductor/dielectric interface on charge noise is well-known in the community, we believe that the most prominent merit of this paper is to focus the attention also on the epitaxial layers of the stack, in particular on the quantum well thickness. We see heterostructure A and B as preliminary results leading to heterostructure C, the main focus of the paper. In view of these considerations, we prefer to keep the title as is, however we leave the final choice to the discretion of the Editor.

1.2 It mentions in the paper that by using the amorphous Si cap can reduce impurities. May I ask which impurities you mean and why using amorphous Si can reduce the impurities?

Our reply: In Ref.[1] we show that passivation of the SiGe surface with amorphous Si results in a more uniform process on a wafer scale compared to an epitaxial Si cap. Moreover, it creates a marked Ge pile-up at the SiGe/SiO_x interface without leaving any residuals of unoxidized Si. The detailed quantum transport analysis in the same paper shows a clear reduction of scattering from remote impurities at the dielectric interface. Although we do not have a clear indication of which kind of impurities are reduced, we can speculate that the exposure of the surface to DSC is possibly beneficial to saturate the surface with H⁺ bonds and obtain very uniform natural oxidation.

1.3 “No structural defects such as misfit dislocation are visible suggesting they are at most scarce” – It is anyway hard to observe misfit dislocation in cross-view TEM since the magnification is too high and the view scale is very limited. Figure 1b is probably around 100 nm x 100 nm.

Our reply: This is a fair point. However since (loosely speaking) this and other new TEM images (such as in the Supplementary) confirm the high quality of the growth, we edited the sentence describing Fig 1c as following:

The $\simeq 5$ nm thick quantum well (Fig. 1c, Supplementary Fig. 1) is uniform, has sharp interfaces to the nearby SiGe, and appears of high crystalline quality.

1.4 This study shows that reducing the quantum well thickness can reduce the charge noise in the quantum dots. Till how thin can the quantum well thickness be reduced?

Our reply: We thank the Reviewer for this question. At some point (which we do not know), further reducing the quantum well thickness could have the negative effect of increasing scattering from the bottom interface, and therefore disorder. In a related study investigating the variance in valley splitting for different quantum well thickness[2], a degradation was observed for well thicknesses < 5 nm ,although in steps of 2nm. For such thin quantum wells both interfaces become relevant and may increase the formation of defects thus potentially also increasing charge noise. Furthermore, considering that the sharpness of the bottom and top quantum well interfaces, as reported in Ref.[3], is characterised by a length scale of $\simeq 0.8$ nm, a quantum well of about 2 nm would probably have a non-negligible Ge concentration inside of it, bringing additional alloy scattering.

Triggered by the Reviewer comment, in our opinion, another pertinent question could be perhaps the following: starting from a quantum well of 5 nm as a baseline, how thick can we make the QW before onset of relaxation and the negative effects on charge noise as shown for heterostructure A? Future experiments with a quantum well thickness in between 5 and 9 nm might provide useful insights to whether a sweet spots for disorder exists between scattering from the bottom interface and onset of misfit dislocations. If this is the case, it will be of great interest to understand if these potential improvements correlate with better charge noise properties. We are starting systematic experiments in this direction, which clearly go beyond the scope of this manuscript.

1.5 There's a study [Lodari et al, Mat. Quant. Tech. (2021)] from the same group on the Ge hole qubit - by increasing the SiGe top buffer layer thickness, the charge noise is reduced. Will it also be the same case by Si quantum well for electron qubit? Can authors give a comment here?

Our reply: We thank the Referee for acknowledging this other study from the group. Indeed, for a given semiconductor/dielectric interface further separating the quantum well should reduce charge noise. However, the lateral dimensions of quantum dots in Si/SiGe are generally smaller than in Ge/SiGe (or GaAs) due to the heavier effective mass for electrons in Si ($0.19m_e$) compared to holes in Ge ($0.05m_e$). This poses a constrain to the depth at which the Si quantum well can be positioned to achieve good electrostatic control of quantum dots with the top gates. For this reason, we envision a fine tuning of the SiGe barrier thickness around 30-40nm rather than a radical shift to much deeper quantum wells, say 50-60nm from the surface.

1.6 Since I am personally not familiar with charge noise characterization – the charge characterization part makes sense to me, however, other experts to review this part is necessary. Therefore, I actually have several questions regards to this part:

a. The measurement conditions are slightly different from sample A to sample B, C – why?

Our reply: We thank the referee for paying attention to the details of this publication. The reason for the change is that when we started measuring heterostructures B and C we wanted to extend the probing range of our experiment from 100 μHz to 10 μHz . This change in measurement range does not change the results of the paper (especially considering the comparison between heterostructure B and C).

Since this information might be useful to readers not familiar with charge noise measurements, we added the following sentence in the method section:

The charge noise measurements conditions have been slightly modified from sample A to sample B, C to extend the probed frequency range from 100 μHz to 10 μHz .

b. Maybe add some example of the ISD sampling in the supplementary data – Plot of time-resolved ISD and related f-dependent SI.

Our reply: We thank the referee for this suggestion and we added in the revised Supplementary Fig. 2 the time-resolved $I_S D$ measurements corresponding to the frequency-dependent S_I shown in the original Supplementary Fig. 3a.

c. The author said, “For each charge noise measurement at a given VP we acquire 60 s (heterostructure A) or 600 s (heterostructures B, C) long traces of ISD and split them into 10 (heterostructure A) or 15 windows (heterostructures B, C). We obtain the current noise spectrum SI by averaging over the 10 (15) windows the discrete Fourier transform of the segments (Methods).” What’s the meaning of windows here?

Our reply: Each window is a time trace of 6 seconds (for heterostructure A) or 40 seconds (for heterostructures B, C). We average subsequently over all the time traces to increase the signal-to-noise ratio. As a consequence we obtain a cleaner Fourier transformed signal. This method (originally proposed by Bartlett[4]) is routinely used for spectral density estimation. A similar method is used in Ref.[5].

d. The equation $\log S_e = -\alpha \log f + \beta V_p + \gamma$. May I ask the source? Maybe can also be added into supplementary materials

We came up with this equation as an extension of the well-known power-law dependence of the noise spectrum with frequency. With this extension we include empirically the observed dependence of the noise spectrum with V_P . We follow the suggestion of the Reviewer and now include the discussion below in the Supplementary Information Section:

We formulated the equation for the plane in Fig. 2d

$$\log S_\epsilon = -\alpha \log f + \beta V_P + \gamma \quad (1)$$

to capture the experimental dependence of $S_\epsilon(f, V_P)$ observed when S_ϵ and f are plotted on a log scale. Eq. 1 expands the known expression describing the low-frequency dependence of the charge noise spectrum [5, 6]

$$S_\epsilon = \frac{A}{f^\alpha}, \quad (2)$$

where A is a constant prefactor describing the noise strength evaluated at 1 Hz. We now multiply a phenomenological prefactor $e^{\beta V_P}$, that describes the reduction in susceptibility of the quantum dot to electric noise as a function of plunger voltage V_P [7]. This yields the expression

$$S_\epsilon = \frac{A}{f^\alpha} e^{\beta V_P}. \quad (3)$$

By taking the logarithm of Eq. 3 and defining $\gamma = \log(A)$ as the unscreened noise, we arrive at the expression Eq. 1 in the main text.

Furthermore, thanks to comments from Reviewer 2, we now report β in the text and Fig. 3 with the correct units, which are mV^{-1} .

1.7) Citation [50] is about superconducting qubits, and I didn't find the equation listed in the citation. There are some similar equations in citation [50] but not the same. Instead, the equation was found in [Bermeister, APL (2013)]. Again, since I am not an expert in charge noise, maybe it's my mistake.

Our reply: We thank the Reviewer for raising this point. The Reviewer is correct, our referenced paper considers the general case and the formula can be derived from the equations. However, the explicit expression used in our work is not present. To the best of our knowledge the expression we used appeared first in Ref. [8] for the general case of a qubit and indeed in Ref. [9] for a spin-qubit. We have modified our manuscript to reflect this.

Finally, the Reviewer makes the following comment about strain:

– I think the strain relaxation measured by Raman shift needs further discussion also. Raman shift can differ from the same sample dependent on the measurement

position. Therefore, only one plot from one measurement point from each sample to say one sample is more relaxed than another one makes not much sense. Besides, I don't think even the Si layer is beyond critical thickness, and there's partial relaxation, but the relaxation can be up to 0.4%. According to the calculation from Matthews-Blakeslee paper, the relaxation of Si with Misfit dislocation spacing of around 1 micron is around 0.02%, that is way smaller than the value observed in this paper.

Our reply: We thank the Reviewer for raising this point about Raman spectroscopy, which triggered further in-depth discussions and considerations. We first recall that the purpose of including Raman characterisation in this paper was to provide a reasonable and plausible argument for the presence (suppression) of misfit dislocation in heterostructure B (C), based on strain relaxation. Perhaps the best choice would have been to directly visualize the misfit dislocations with electron channeling contrast imaging (such as in Ref. [10]). Also after consultation with other Raman experts, we agree with the Reviewer that, indeed, showing only one spectrum might not yield firm conclusions when comparing our two heterostructures, *especially* considering the presence of the typical cross-hatch pattern arising from the dense array of misfit dislocations in the lower layers of the SiGe virtual substrate.

The cross-hatch pattern is characterised by a wavelength of about 1 μm (see for example Supplementary Fig. 14c of Ref. [3]) and is also a fingerprint of strain and composition fluctuations (see for example [11, 12]). Since the Raman spot size in our characterisation was also about 1 μm , in order to make a firm comparison between different heterostructures one should perform high-resolution spatially-resolved scanning Raman spectroscopy and compare distributions/bandwidth of strain over the sample rather than single spectra (see for example supplementary information of doi.org/10.1021/acsami.2c17395 for Ge/SiGe quantum wells).

While we feel that this extensive characterisation goes beyond the scope of our manuscript, it is for sure an interesting avenue to pursue in future experiments. However, given the clear and convincing measurements of misfit dislocations by electron channeling contrast imaging reported in doi.org/10.1063/5.0101753 (published only a few weeks before we submitted our own paper) on heterostructures very similar to ours, we deem our plausibility argument for the presence (absence) of misfits in heterostructure B (C) reasonable. Therefore we prefer to remove the strain characterisation from the Supplementary Section and the text in the manuscript referring to it. Accordingly, we have rearranged and edited the description of Fig. 1a as following:

In light of recent morphological characterisation by electron channeling contrast imaging of Si/SiGe heterostructures with similar quantum well thickness and SiGe chemical composition[10], we expect misfit dislocation segments in heterostructure B because the quantum well approaches the Matthews-Blakeslee critical thickness. Due to the much thinner quantum well, instead, the epitaxial planes may adapt to the SiGe buffer much better in heterostructure C than in heterostructure B, meaning that misfit dislocations are, in

principle, suppressed[10].

2. RESPONSE TO REVIEWER 2

We thank Reviewer 2 for welcoming our study of charge noise in Si/SiGe quantum dots and for strongly recommending publication of the manuscript in Nature Communications. We thank the Reviewer also for raising the the few technical comments, that we address below.

1) On pg 3, it would be helpful to state why the specific numbers given for the noise trace lengths and number of windows were chosen, and in particular why they are not chosen to be the same for all heterostructures.

Our reply: Those numbers were chosen to achieve an appropriate frequency for the measurement. As explained our reply to point 1.6 of Reviewer 1, we changed the parameters from when we started measuring heterostructures B and C because we wanted to extend the low end range of our experiment from 100 μHz to 10 μHz . This change in measurement range does not change the results of the paper. We added a note in the methods to clarify

2) On pg 4, the units of β do not make sense to me. It cannot have the same units as S/V_P .

Our reply: We thank the Reviewer for this point. Indeed, this is a mistake that we fixed. As explained in our reply to point 1.5d of Reviewer 1, the correct units for β are mV^{-1} .

3) Fig. 2d, from heterostructure C, is said to be fit by the plane with $\alpha = 0.84$, but in Fig. 3c this value seems to be at the extreme lower end of the distribution. Did the data for Fig. 2d come from an atypical H-FET?

Our reply: We did not choose a specific device based on it's position within the distribution. Indeed, it happens that this device is in the lower end of the distributions for alpha, but still within the interquartile range in Fig .3c. Importantly, this device is not an outlier and also the V_P range of 250–350 mV (as seen in Fig 2e) is in line with the other devices from this heterostructure (as seen in Supplementary Fig. 4a). Based on these considerations, we deem our choice reasonable and this H-FET not an atypical device.

4) On pg 5, the authors write that "the larger spread in α in heterostructure B and C implies that deviations from 1/f behaviour become more frequent, possibly originating from a non-uniform distribution of TLF or from one low frequency TLF in the surrounding environment..." While I agree that a nonuniform distribution of TLF activation energies would cause deviations from 1/f behavior, it is not clear to me why one should expect that the remaining dislocations in B would have

a less uniform distribution of activation energies than the interfacial impurities. Do the authors have an idea why this might be? Furthermore, if one simply has a dominant low-frequency TLF nearby, the power spectrum would be Lorentzian and would certainly not look like Fig 2 over nearly three decades of frequency. That is, unless that TLF also experiences slow temperature fluctuations, in which case Ref. 42 has shown that a single TLF *does* produce $1/f$ noise (intriguingly, with α also around 0.8). The authors should clarify this.

Our reply: We thank the Reviewer for this comment, that triggered further discussion about the interpretation of this particular trend, i.e. how the spread in α changes going from heterostructure A to B and C. To answer the question from the Reviewer: we do not know if the activation energies of charge noise sources associated with misfit dislocations are non-uniformly distributed. What we know is that misfit dislocations due to partial relaxation of the strained Si are highly localized defects. For example, see electron channeling contrast imaging in Fig. 4a of Ref [10] that visualizes misfit dislocations in a heterostructure with composition and quantum well thickness very similar to our heterostructure A and B. Even more striking is Fig. 3c in Ref. [12], where a misfit dislocation nearby 2 (functional) qubit devices in Ge/SiGe is shown. Consequently, the electronic states that misfit dislocations cause in their vicinity, and the associated charge noise sources, are inhomogeneously distributed across the wafer.

We thank the Reviewer for the remark about $1/f$ like noise and the possibility to observe such even with local fluctuators according to ref 42. We have added a supplementary figure showing for the quantum dot in Fig.2 (heterostructure C) all noise spectra, indeed showing a $1/f$ like behaviour. Our understanding is that in heterostructure A noise from the TLF at the remote oxide interface is possibly dominating over noise associated with misfit dislocations. In heterostructure B, as the oxide interface gets better, but misfit dislocations are the same as in heterostructure A, the contribution to noise from more localized sources appears as an increased spread in α . In heterostructure C, noise from misfit dislocation is also greatly reduced, and we are left with even larger fluctuations of α . Indeed, and we thank the Reviewer for pointing this out, based on the results in Ref [13] these nearby fluctuators could give rise to noise spectra similar to the ones we see.

We edited the relevant paragraph in the manuscript to capture these considerations:

The narrow distribution of α in heterostructure A points to charge noise being dominated from many TLFs possibly located at the low quality semiconductor/dielectric interface and above, albeit other sources of charge noise in the surrounding environment of the quantum dot may be present, such as highly localized misfit dislocations arising from partial strain relaxation in the quantum well or other nearby fluctuators. With a better semiconductor/dielectric interface, the effect of these other nearby fluctuators emerges in heterostructure B and C as a larger spread of the frequency exponent α , indicating a nonuniform distribution of activation energies according to the Dutta-Horn model[14]. Yet, the noise spectra still follow a $1/f$ -like behavior (Supplementary Fig. 3), suggesting that TLFs also experience slow temperature fluctuations[13].

5) On pg 5, I agree that it makes sense to interpret the decreasing trend in $|\beta|$ as a reduction in the effectiveness of screening. That is consistent with the charge noise being increasingly dominated by nearby TLFs rather than remote scatterers. However, I do not know if that is what the authors have in mind when they mention a smaller "TLF-per-volume" ratio, and this phrase should be clarified.

Our reply: We thank the Reviewer for spotting this unclear wording. Indeed, the Reviewer's understanding is correct. By smaller TLF per volume ratio" we simply meant a "smaller density of TLFs". Since this is already said at the beginning of the sentence, we simply removed this wording.

As the impurity density decreases from heterostructure A to B and C, charge noise is less affected by an increasing V_P , since screening of electrical noise through adding electrons to the charge sensor becomes less effective.

3. RESPONSE TO REVIEWER 3

We thank the Reviewer for appreciating our work and believing that the paper is a strong candidate for publication. The Reviewer brings forward and encourage us to consider opportunities for clarity, discussion, and impact. We welcome these points that we address below.

3.1 In the example determination of $S_{\epsilon, min}$ presented in Fig. 2 and S3, the conversion from S_I to S_{ϵ} depends on the lever arm and the slope of the Coulomb blockade peak ($|dV/dp|$), as defined in Eq. 3. Do the authors use a single value of the lever arm/slope for each device, or is the value uniquely determined for each peak? If the former, then some justification and discussion of this choice is appropriate. If the latter, I would welcome an additional supplemental figure which demonstrated for the case presented in Fig 2 the values of lever arm and slope found – not just for assessing the conversion of data to charge noise, but also for consideration in subsequent discussion of screening, etc. I would imagine this as similar to Fig S4, which shows ranges of other operating parameters.

Our reply: We thank the Reviewer for this comment. Indeed, the lever arm value used for converting S_I to S_{ϵ} is the value uniquely determined for each Coulomb peak. Caption for Fig. 2c is modified accordingly (see our reply to point 3.2 below). We also corrected for some typos in writing the conversion equation in the Method section:

To convert the current noise spectrum to a charge noise spectrum, we use the formula [5]

$$S_{\epsilon} = \frac{a^2 S_I}{|dI/dV_P|^2} \quad (4)$$

where a is the lever arm and $|dI/dV_P|^2$ is the slope of the Coulomb peak **at the plunger voltage used to acquire the time trace.**

As requested by the Reviewer, we added a new Supplemental Figure showing lever arms and the Coulomb blockade slopes a function of V_P for the device featured in Fig 2. We observe a clear anticorrelation between lever arm and plunger gate voltage, implying a progressive change in the quantum dot size and its coupling to the gate. This is also followed by a correlation between slope $|dI/dV_P|$ and V_P .

The following changes in the main text reflect these considerations:

We convert S_I to a charge noise spectrum S_ϵ using, **for each measurement at a given V_P , the unique lever arm from the corresponding Coulomb diamonds and slope of the Coulomb peak to take into account a possible deformation of the charge sensor with the increasing electron number (inset Fig. 2b, Methods, and Supplementary Fig.5).**

3.2 Continuing from Fig 2d, where the derived S_ϵ data for various CB peaks are plotted along with the “plane” used to capture the character of the device, where is $S_{\epsilon, min}$ in this plot (Fig 2d) as determined by the authors? Please add a special symbol to explicitly show the $S_{\epsilon, min}$ location in this plot. Additionally, if the same units were used in Figs 2 c – e, this would be much more helpful to the reader. Similarly for showing $S_{\epsilon, min}$ in Fig 2e and connecting to the data in Fig S4.

Our reply: We thank the Reviewer for these further comments/questions on Fig. 2. By addressing them, as we describe below, we believe we have improved the overall clarity of Fig. 2, and consequently of the manuscript. The changes follow the narrative of the manuscript, where we first (Fig. 2c) show a generic noise spectrum at with an approximate $1/f$ trend, then plot in Fig. 2d all the spectra at different V_P (along with the fitted “plane”), and finally, take a line-cut at 1 Hz (Fig. 2e) to highlight the V_P dependence and extract the minimum charge noise $S_{\epsilon, min}$.

Corresponding changes to figure panels:

- Fig 2c: We removed the fit (this may be confusing, since it is done in **d**) and added a $1/f$ trendline instead.
- Fig 2d, e: As requested by the Reviewer, we added a symbol (black circle) to explicitly show the $S_{\epsilon, min}$ location in these plot. Accordingly, in the main text describing this section we refer to $S_{\epsilon, min}$, rather than to its square root.
- Fig 2e. As requested by the Reviewer, we plotted S_ϵ in the same units as in **c,d** (similar change in Fig. S4).

The caption for Fig.2c-e has been edited as following:

c Charge noise spectrum S_ϵ measured at the Coulomb peak at $V_P \simeq 360.3$ mV in **b** and extracted using the lever arm from the corresponding Coulomb diamond. The black trendline is proportional to $1/f$. **d** S_ϵ for the same device in **b**, plotted in 3D as a function of f and V_P . The dark gray plane is a fit through the datasets, i.e. the collection of noise spectra as in **c** measured at different V_P and each obtained using a unique lever arm from the corresponding Coulomb diamond. **e** Line cut through the data in **d** at $f = 1$ Hz, showing the experimental noise S_ϵ (colored dots) and fit (dark grey line). The black circled data point (also in **d**) marks the minimum charge noise measured for this specific device ($S_{\epsilon,min}$) at $f = 1$ Hz.

In the following Fig. 3e we decided to keep our original notation ($S_{\epsilon,min}^{1/2}$) so that charge noise is expressed in units of $\mu\text{eV}/\sqrt{\text{Hz}}$ and may be easily compared to the literature values:

Finally, Fig. 3e shows the distributions for $S_{\epsilon,min}^{1/2}$, the minimum charge noise at 1 Hz in units of $\mu\text{eV}/\sqrt{\text{Hz}}$ upon varying V_P .

3.3 While I recognize the main experimental part leverages comparison based on parameters for the fit function (what happens with gamma ?), the theoretical extrapolation to T_{2*} and F are determined entirely by $S_{E,min}$. As presented, the data, extrapolations, and eventual conclusions can only be qualitatively compared since it is not clear how $S_{E,min}$ is determined by the authors, and why that determination is a robust means of making comparisons between the devices, which have some variation in tuning, etc. Particularly for the choices in the theoretical section, and the choices of things like the high and low frequency cutoffs (which don't overlap with the data at all in one case?) additional discussion about the robustness for intercomparison and whether the $S_{E,min}$ is robust if the same device was tuned up multiple times would help establish more confidence in this approach. Why should the reader trust $S_{E,min}$?

Our reply: We thank the Reviewer for this critical and insightful comment. We first answer the question regarding γ . We have added a phenomenological derivation of our fitting function (see reply to Reviewer 1, point 1.6d and new Supplementary note). In this setup the constant γ only denotes the PSD at 1 Hz and $V_P = 0$ and contains no information. However, with knowledge of $V_{P,1e}$ (single electron regime), γ will denote the unscreened PSD at 1 Hz.

To answer the rest of the comment, the Reviewer is right, the theoretical extrapolation to dephasing times and fidelities which we use for cross platform comparison bases solely on the minimal observed PSD at 1 Hz $S_{E,min}$ and on the noise exponent α . Our choice for $S_{E,min}$ was made on two observations, taking into account also the values for $S_{E,max}$. $S_{E,max}$, now reported in Supplementary Fig. 4, is the maximum charge noise measured for a given dot in a heterostructure upon varying V_P within the available range. First, the trend observed for

$S_{E,max}$ confirms the trend observed for $S_{E,min}$: charge noise improves as we move from A to B and to C. Therefore this trend is not limited to the specific choice of $S_{E,min}$ as a metric. Second, comparing the spread in $S_{E,min}$ to the spread in $S_{E,max}$, we see typically smaller variations of $S_{E,min}$ between different devices for a given heterostructure (see new figure in Supplement). $S_{E,max}$, instead, varies stronger between different devices as it is more affected by device-specific effects such as geometry of wave-function, screening, and the exact electron number on the island. Because we do not know the exact electron number, we believe that $S_{E,min}$ is more suited to compare between the different heterostructures.

While we haven't checked if $S_{E,min}$ is robust if the same device was tuned up multiple times, we feel confident that $S_{E,min}$ is a good metric for intercomparison of charge noise from heterostructure A to B and C also because we do measure $S_{E,min}$ across multiple devices which inevitably will have some differences in tune-ups since this depends on the local disorder potential landscape. The merit of our analysis is to draw conclusions about charge noise in the different heterostructures based on distributions of metrics, rather than a single device.

We have performed the following edits to the manuscript to capture these considerations:

We find in heterostructure C an almost order of magnitude reduction in mean $S_{\epsilon,min}^{1/2}$ to $0.29 \pm 0.02 \mu\text{eV}/\sqrt{\text{Hz}}$. **This trend is confirmed by plotting the distributions of maximum charge noise at 1 Hz upon varying V_P (Supplementary Fig. 4).**

Furthermore, for the sake of transparency, we have previously indicated in the caption of Fig. 4 that we use the distribution of experimental $S_{E,min}$ as input parameter. We realize that this information should be given more prominently in the main text. Therefore, we have now added a remark in the theory section to explain that, as a consequence of this choice, our calculations present a best case scenario (or upper bound) to dephasing times.

Figure 4a shows the computed dephasing times of charge qubits (circle) and spin qubits (star) Furthermore, for three heterostructures. **These calculations represent a best case scenario, since we use the distribution of measured $S_{\epsilon,min}$ from Fig. 3. as input parameter for each heterostructure.**

Finally, we thank the Reviewer for the comment about the cutoff frequencies. The Reviewer is right, that our simulation of the dephasing time for the charge qubit the simulated frequency bandwidth is outside the measured frequency bandwidth. This choice was made to provide a better comparison to the experiments as charge qubits are operated on different time-scales than spin-qubits. We also want to show the influence of the noise exponent on the dephasing times as an identical bandwidth for the simulated spin and charge qubit dephasing times would only result in a constant shift due to the prefactor $\left| \frac{\partial \mathcal{E}}{\partial \mu} \right|$. Nevertheless, in light of the Reviewer

comment, we have decreased the cutoff frequency f_{min} to 1.6 mHz for the calculations of the charge qubit dephasing time. There is no major change to the result obtained, and Fig. 4a has been updated accordingly. For increased transparency, we also indicate the frequency cutoff frequencies by adding the following sentence to the caption of Fig. 4.

Eq. 1 was used to compute T_2^* as a function of S_ϵ and α from Fig. 3 **with frequency cutoffs** $(f_{min}, f_{max}) = (1.6 \text{ mHz}, 33 \text{ GHz})$ and $(f_{min}, f_{max}) = (1.6 \text{ mHz}, 10 \text{ kHz})$.

As a final comment, the Reviewer states that: **Finally, in the theoretical section and supplemental, additional attention to wordsmithing and imprecise language could help with clarity.**

Our reply: We thank the Reviewer for this last final comment performed minor language changes to these sections to make them more clear.

Furthermore, we have updated the device fabrication method section with more information:

Device fabrication. The fabrication process for Hall-bar shaped heterostructure field effect transistors (H-FETs) involves: reactive ion etching of mesa-trench to isolate the two-dimensional electron gas; P-ion implantation and activation by rapid thermal annealing at 700 °C; atomic layer deposition of a 10-nm-thick Al_2O_3 gate oxide ; **deposition of thick dielectric pads to protect gate oxide during subsequent wire bonding step**; sputtering of Al gate; electron beam evaporation of Ti:Pt to create ohmic contacts to the two-dimensional electron gas via doped areas. All patterning is done by optical lithography. Double quantum dot devices are fabricated on wafer coupons from the same H-FET fabrication run and share the process steps listed above. Double-quantum dot devices feature a single layer gate metallization and further require electron beam lithography, evaporation of **Al (27 nm)** or Ti:Pt (3:17 nm) thin film metal gate, lift-off, **ALD of a 5 nm thick Al_2O_3 insulating layer**, and a global top-gate.

-
- [1] D. Degli Esposti, B. Paquelet Wuetz, V. Fezzi, M. Lodari, A. Sammak, and G. Scappucci, Applied Physics Letters **120**, 184003 (2022).
 - [2] E. H. Chen, K. Raach, A. Pan, A. A. Kiselev, E. Acuna, J. Z. Blumoff, T. Brecht, M. D. Choi, W. Ha, D. R. Hulbert, M. P. Jura, T. E. Keating, R. Noah, B. Sun, B. J. Thomas, M. G. Borselli, C. Jackson, M. T. Rakher, and R. S. Ross, Physical Review Applied **15**, 044033 (2021).
 - [3] B. Paquelet Wuetz, M. P. Losert, S. Koelling, L. E. A. Stehouwer, A.-M. J. Zwerver, S. G. J. Philips, M. T. Mądzik, X. Xue, G. Zheng, M. Lodari, S. V. Amitonov, N. Samkharadze, A. Sammak, L. M. K. Vandersypen, R. Rahman, S. N. Coppersmith, O. Moutanabbir, M. Friesen, and G. Scappucci, Nature Communications **13**, 7730 (2022).

- [4] M. S. Bartlett, *Nature* **161**, 686 (1948).
- [5] E. J. Connors, J. J. Nelson, H. Qiao, L. F. Edge, and J. M. Nichol, *Physical Review B* **100**, 165305 (2019).
- [6] M. Lodari, N. W. Hendrickx, W. I. L. Lawrie, T.-K. Hsiao, L. M. K. Vandersypen, A. Sammak, M. Veldhorst, and G. Scappucci, *Materials for Quantum Technology* **1**, 11002 (2021).
- [7] C. Spence, B. Cardoso-Paz, V. Michal, E. Chanrion, D. J. Niegemann, B. Jadot, P.-A. Mortemousque, B. Klemt, V. Thiney, B. Bertrand, L. Hutin, C. Bäuerle, F. Balestro, M. Vinet, Y.-M. Niquet, T. Meunier, and M. Urdampilleta, “Probing charge noise in few electron CMOS quantum dots,” (2022), arXiv:2209.01853 [cond-mat].
- [8] A. Shnirman, Y. Makhlin, and G. Schön, *Physica Scripta* **2002**, 147 (2002).
- [9] A. Bermeister, D. Keith, and D. Culcer, *Applied Physics Letters* **105**, 192102 (2014).
- [10] Y. Liu, K.-P. Gradwohl, C.-H. Lu, T. Remmele, Y. Yamamoto, M. H. Zoellner, T. Schroeder, T. Boeck, H. Amari, C. Richter, and M. Albrecht, *Journal of Applied Physics* **132**, 085302 (2022).
- [11] M. H. Zoellner, M.-I. Richard, G. A. Chahine, P. Zaumseil, C. Reich, G. Capellini, F. Montalenti, A. Marzegalli, Y.-H. Xie, T. U. Schüllli, M. Häberlen, P. Storck, and T. Schroeder, *ACS Applied Materials & Interfaces* **7**, 9031 (2015).
- [12] C. Corley-Wiciak, C. Richter, M. H. Zoellner, I. Zaitsev, C. L. Manganelli, E. Zatterin, T. U. Schüllli, A. A. Corley-Wiciak, J. Katzer, F. Reichmann, W. M. Klesse, N. W. Hendrickx, A. Sammak, M. Veldhorst, G. Scappucci, M. Virgilio, and G. Capellini, *ACS Applied Materials & Interfaces*, acsami.2c17395 (2023).
- [13] S. Ahn, S. D. Sarma, and J. P. Kestner, *Physical Review B* **103**, L041304 (2021).
- [14] P. Dutta, P. Dimon, and P. M. Horn, *Physical Review Letters* **43**, 646 (1979).
- [15] X. Xue, M. Russ, N. Samkharadze, B. Undseth, A. Sammak, G. Scappucci, and L. M. K. Vandersypen, *Nature* **601**, 343 (2022).
- [16] S. G. J. Philips, M. T. Mądzik, S. V. Amitonov, S. L. de Snoo, M. Russ, N. Kalhor, C. Volk, W. I. L. Lawrie, D. Brousse, L. Tryputen, B. P. Wuetz, A. Sammak, M. Veldhorst, G. Scappucci, and L. M. K. Vandersypen, *Nature* **609**, 919 (2022).

Figure R1. Comparison, for illustration purposes, of charge noise measurements spectra under different conditions. **a** Coulomb peak with large derivative dI_{SD}/dV_P and **b** time-resolved I_{SD} measured at the flank of the Coulomb peak (dot in **a**). Measurements are from a device from heterostructure C. **c** Coulomb peak with smaller derivative dI_{SD}/dV_P and **d** time-resolved I_{SD} measured at the flank of the Coulomb peak (dot in **c**). Measurements are from a device from heterostructure B. **e** Coulomb blockade and **d** time-resolved I_{SD} measured on a test device from heterostructure B, indicative of the noise floor of our measurement setup. The time traces in **b,d,f** show a consistent decrease in the noise bandwidth going from the most sensitive ($\Delta I_{SD} \simeq 50$ pA in **b**) to the less sensitive ($\Delta I_{SD} \simeq 10$ pA in **f**) configuration. **g** Comparison of the current noise spectrum under different sensitivity conditions. Purple (high sensitivity), cyan (low-sensitivity), and lemon (noise floor) curves shows $S_I(f)$ obtained from measurements in **b**, **d**, and **f**, respectively. Lemon and cyan curves show a broad interference peak at 150 Hz, as well as a flattening out of the curve at ≈ 40 Hz. **h** Charge noise measurement of heterostructure A with an interference peak at 9 Hz arising from the measurement module. In **i** we remove the interference peak from the analysis. **l,m** Charge noise of a device from heterostructure B and C, respectively, measured with a different measurement module compared to **l** showing no interference peak.

Figure R2. **a** False colored SEM-image of a double quantum dot system with a nearby charge sensor. Charge noise is measured in the multi-electron quantum dot defined by accumulation gates SDRAcc and SDRAcc (blue), plunger P (blue), with the current going along the black arrow. In these experiments, the gates defining the double quantum dot (red) are used as screening gates. There is an additional global top gate (not shown) to facilitate charge accumulation when needed. **b** Source-drain current I_{SD} through a charge sensor device fabricated on heterostructure C against the plunger gate voltage V_P . Colored dots mark the position of the flank of the Coulomb peak where charge noise measurements are performed. The inset shows Coulomb diamonds from the same device, plotted as the differential of the current dI/dV as a function of V_P and the source drain bias V_{SD} . **c** Charge noise spectrum S_ϵ measured at the Coulomb peak at $V_P \simeq 360.3$ mV in **b** and extracted using the corresponding lever arm from Coulomb diamonds. The black trendline is proportional to $1/f$. **d** S_ϵ for the same device in **b**, plotted in 3D as a function of f and V_P . The dark gray plane is a fit through the datasets, i.e. the collection of noise spectra as in **c** measured at different V_P . **e** Line cut through the data in **d** at $f = 1$ Hz, showing the experimental noise S_ϵ (colored dots) and fit (grey line). The black circled data point (also in **d**) marks the minimum charge noise measured for this specific device ($S_{\epsilon,min}$) at $f = 1$ Hz.

Figure R3. Charge noise spectra $S_c(f)$ at different plunger gate voltage V_P from a quantum dot from heterostructure C. The same data is plotted in Fig. 2d in three dimensions. The black trendline shows a $1/f$ dependence.

Figure R4. **a** Charge noise $S_e^{1/2}$ at 1 Hz as a function of the plunger gate voltage V_P for all measured devices of heterostructure A (red), B (blue), and C (green). Circles and diamonds highlight, respectively, the minimum ($S_{e,min}$) and maximum ($S_{e,max}$) charge noise at 1 Hz for each device upon varying V_P . For a given heterostructure, these $S_{e,min}$ and $S_{e,max}$ values build up the distributions plotted, respectively, in **b** (Fig. 2e main text) and **c**. The trend of charge noise improvement from A to B and C is observed both for $S_{E,min}$ and $S_{E,max}$. $S_{E,min}$ varies less than $S_{E,max}$ between different devices for a given heterostructure since $S_{E,max}$ is more affected by device-specific effects such as geometry of wave-function, screening, and the exact electron number on the island. Because we do not know the exact electron number, we believe that $S_{E,min}$ is more suited to compare between the different heterostructures. **d-i** Distributions of the operation gate voltages of the plunger, SDLAcc, SDLB, SDRB, SDRAcc, and screening gates, respectively (see Fig. 1f in the main text) for heterostructure A (red, 4 devices measured), B (blue, 8 devices measured), and C (green, 5 devices measured). With the exception of gate SDLB, all operation voltages of the charge sensor are highest in heterostructure A and lowest in heterostructure C with a difference of up to 600 mV. Note that a global screening gate is only used for the operation of heterostructure A. Quartile box plots, mode (horizontal line), means (diamonds), 99% confidence intervals of the mean (dashed whiskers), and outliers (circles) are shown.

Figure R5. **a** Differential conductance (dI/dV) showing representative Coulomb blockade diamonds as a function of the source-drain voltage (V_{SD}) and plunger gate voltage (V_P) for heterostructure C. We derive the two slopes m_S and m_D on both sides of each Coulomb diamond. Using the equation $a = \left| \frac{m_S m_D}{m_S - m_D} \right|$, we extract a lever arm of $a = 0.12$ eV/V for the Coulomb peak at $V_P \approx 308$ mV, where we indicate m_S and m_D with magenta lines. The dashed line indicates the source-drain voltage ($V_{SD} = 150$ (μV)) used for the charge noise measurements. **b** Lever arm and **c** slope at the flank of the Coulomb peak for the peaks reported in **a**. We calculate the Pearson correlation coefficient (r), measuring the linear correlation between the two parameters. It varies between -1 and 1, with 0 implying no correlation. We also calculate the p-value of the null hypothesis, i.e., $r = 0$. We remember that the p-value indicates the probability of an uncorrelated system producing datasets that have a Pearson correlation at least as extreme as the one computed from these datasets. We remember that a p-value greater than 0.05 is considered not statistically relevant.

Figure R6. **Calculated dephasing times and infidelity.** **a** Computed dephasing times T_2^* of a charge qubit (circle) and of a spin-qubit (star) using $S_{\epsilon, \min}$ from heterostructure A (red), B (blue), C (green). Eq. 1 was used to compute T_2^* as a function of S_{ϵ} and α from Fig. 3 **with frequency cutoffs** $(f_{\min}, f_{\max}) = (1.6 \text{ mHz}, 33 \text{ GHz})$ and $(f_{\min}, f_{\max}) = (1.6 \text{ mHz}, 10 \text{ kHz})$. Literature values (squares) are taken from Refs. [15, 16]. **b** Simulated infidelity of a CZ-gate between two spin qubits following the Ref. [15] using S_{ϵ} and α from heterostructure A (red), B (blue), C (green) in Fig. 3 as input for barrier fluctuations.

REVIEWERS' COMMENTS

Reviewer #1 (Remarks to the Author):

Thanks for the careful responses from the authors to all the minor points raised. The questions I proposed are all well answered. I would suggest the authors further add the discussion of “1.4 This study shows that reducing the quantum well thickness can reduce the charge noise in the quantum dots. Till how thin can the quantum well thickness be reduced?” into the manuscript when this discussion is not confidential.

Reviewer #2 (Remarks to the Author):

The authors have satisfactorily addressed my comments. I have already recommended publication in Nature Communications.

Reviewer #3 (Remarks to the Author):

The authors have rigorously and conscientiously responded to the comments of the reviewers and made numerous revisions to the manuscript to improve clarity and completeness on what was already a very comprehensive presentation of the work. I have no reservations recommending the article for publication.

Response to Reviewer 1,

We thank for approving the manuscript for publication. We have accommodated the last request about including the discussion of your point 1.4 by adding the following sentence when discussing Fig3a,b:

We speculate that further reducing the quantum well thickness could increase surface roughness scattering from the bottom interface, and therefore disorder. Instead, fine-tuning the quantum well thickness between 5 nm and 9 nm might minimize surface roughness scattering whilst still avoiding the formation of misfit dislocations.